# Silymarin Inhibits Glutamate Release and Prevents against Kainic Acid-Induced Excitotoxic Injury in Rats

**DOI:** 10.3390/biomedicines8110486

**Published:** 2020-11-09

**Authors:** Cheng-Wei Lu, Tzu-Yu Lin, Kuan-Ming Chiu, Ming-Yi Lee, Jih-Hsin Huang, Su-Jane Wang

**Affiliations:** 1Department of Anesthesiology, Far-Eastern Memorial Hospital, New Taipei 22060, Taiwan; drluchengwei@gmail.com (C.-W.L.); drlin1971@gmail.com (T.-Y.L.); 2Department of Mechanical Engineering, Yuan Ze University, Taoyuan 32003, Taiwan; 3Division of Cardiovascular Surgery, Cardiovascular Center, Far-Eastern Memorial Hospital, New Taipei 22060, Taiwan; chiu9101018@gmail.com (K.-M.C.); mingyi.lee@gmail.com (M.-Y.L.); 8132va@gmail.com (J.-H.H.); 4Department of Nursing, Oriental Institute of Technology, New Taipei 22060, Taiwan; 5Department of Photonics Engineering, Yuan Ze University, Taoyuan 32003, Taiwan; 6School of Medicine, Fu Jen Catholic University, New Taipei 24205, Taiwan; 7Research Center for Chinese Herbal Medicine, College of Human Ecology, Chang Gung University of Science and Technology, Taoyuan 33303, Taiwan

**Keywords:** silymarin, glutamate release, synaptosomes, voltage-dependent Ca^2+^ channels, ERK1/2, neuroprotection

## Abstract

Silymarin, a polyphenoic flavonoid derived from the seeds of milk thistle (*Silybum marianum*), exhibits neuroprotective effects. In this study, we used a model of rat cerebrocortical synaptosomes to investigate whether silymarin affects the release of glutamate, an essential neurotransmitter involved in excitotoxicity. Its possible neuroprotective effect on a rat model of kainic acid (KA)-induced excitotoxicity was also investigated. In rat cortical synaptosomes, silymarin reduced glutamate release and calcium elevation evoked by the K^+^ channel blocker 4-aminopyridine but did not affect glutamate release caused by the Na^+^ channel activator veratridine or the synaptosomal membrane potential. Decreased glutamate release by silymarin was prevented by removal of extracellular calcium and blocking of N- and P/Q-type Ca^2+^ channel or extracellular signal-regulated kinase 1/2 (ERK1/2) but not by blocking of intracellular Ca^2+^ release. Immunoblotting assay results revealed that silymarin reduced 4-aminopyridine-induced phosphorylation of ERK1/2. Moreover, systemic treatment of rats with silymarin (50 or 100 mg/kg) 30 min before systemic KA (15 mg/kg) administration attenuated KA-induced seizures, glutamate concentration elevation, neuronal damage, glial activation, and heat shock protein 70 expression as well as upregulated KA-induced decrease in Akt phosphorylation in the rat hippocampus. Taken together, the present study demonstrated that silymarin depressed synaptosomal glutamate release by suppressing voltage-dependent Ca^2+^ entry and ERK1/2 activity and effectively prevented KA-induced in vivo excitotoxicity.

## 1. Introduction

Glutamate is the main excitatory neurotransmitter controlling fundamental physiological processes in the central nervous system (CNS), such as neuronal development, synaptogenesis, and synaptic plasticity [1,2]. Excessive release of glutamate causes excitotoxicity and results in neurodegeneration, which is observed in several neurological pathologies, including epilepsy, ischemia, intracranial hemorrhage, spinal cord injury, head trauma, and neurodegenerative diseases [3,4,5]. Several clinical neuroprotectants, including riluzole, fluoxetine, and dexmedetomidine, inhibit glutamate release-inhibiting effects [6,7,8]. Thus, suppression of glutamate release is an essential mechanism of neuroprotective agents.

Numerous bioactive natural products have attracted attention for their potential therapeutic effects and safety, including silymarin [9,10]. Silymarin is a polyphenoic flavonoid derived from the seeds of milk thistle (*Silybum marianum*) and has been widely used to treat liver diseases in traditional medicine [11,12]. In addition to hepatoprotection, silymarin has antiallergic, anticancer, antihyperglycemic, and immunoregulatory activities [13,14,15,16]. Silymarin is postulated to penetrate the blood–brain barrier [17] and act on the CNS by suppressing neuroinflammation, attenuating brain damage and ameliorating cognitive deficits in various models of neurologic disorders, such as cerebral ischemia, Alzheimer’s disease, and Parkinson’s disease [18,19,20,21]. However, the effects of silymarin on the synaptic release of glutamate and glutamate excitotoxicity are not elucidated. In the current study, we evaluated whether silymarin affects glutamate release in rat cerebrocortical nerve terminals (synaptosomes) and protects against neurotoxicity induced by the systemic administration of a glutamate analog kainic acid (KA), a well-documented model of glutamate excitotoxicity [22].

## 2. Materials and Methods

### 2.1. Chemical Reagents

Silymarin (5–30 µM, purity >98%, ChemFaces), ethylene glycol bis(β-aminoethyl ether)-N,N,N1,N1-tetraacetic acid (EGTA, 300 µM, Sigma-Aldrich, MO, USA), the vesicular transporter inhibitor bafilomycin A1 (0.1 µM, Tocris, Bristol, UK), endoplasmic reticulum Ca^2+^ release inhibitor dantrolene (10 µM, Tocris, Bristol, UK), mitochondrial Na^+^/Ca^2+^ exchange inhibitor 7-chloro-5-(2-chlorophenyl)-1,5-dihydro-4,1-benzothiazepin-2(3*H*)-one (GP37157, 10 µM, Tocris, Bristol, UK), N- and P/Q-type Ca^2+^ channel inhibitor ω-conotoxin MVIIC (ω-CgTX MVIIC, 4 µM, Alomone lab, Jerusalem, Israel), inositol 1,4,5-trisphosphate (IP_3_)receptor antagonist xestospongin C (1 µM, Tocris, Bristol, UK), protein kinase A (PKA) inhibitor N-[2-(*p*-bromocinnamylamino)ethyl]-5-isoquinolinesulfonamide (H89, 100 µM, Tocris, Bristol, UK), protein kinase C (PKC) inhibitor bisindolylmaleimide I (GF109203X, 10 µM, Tocris, Bristol, UK), mitogen-activated protein kinase (MAPK) inhibitor 2-(2-amino-3-methoxyphenyl)-4H-1-benzopyran-4-one (PD98059, 50 µM, Tocris, Bristol, UK), extracellular signal-regulated kinase 1/2 (ERK1/2) inhibitor FR180204 (10 µM, Tocris, Bristol, UK), Ca^2+^ indicator fura-2-acetoxymethyl ester (Fura-2-AM, 5 µM, Life Technologies, Bengaluru, India), membrane potential-sensitive dye 3′,3′-dipropylthiadicarbocyanine iodide (DiSC_3_(5), 5 µM, Invitrogen, CA, USA), K^+^ channel blocker 4-aminopyridine (1 mM, Sigma-Aldrich, MO, USA), and glutamate analog KA (15 mg/kg, Sigma-Aldrich, MO, USA) were used. The dosages of these agents were chosen based on previous experiments of our group and others [8,23,24,25].

### 2.2. Animals

A total of 57 male Sprague Dawley rats (150–200 g) were used in this study. Twenty rats for in vitro studies and 37 rats for in vivo studies were purchased from BioLASCO Taiwan Co., Ltd. (Taipei City, Taiwan) and bred at the Animal Care Services of Fu Jen Catholic University. All experimental procedures were reviewed and approved by the Institutional Animal Care and Use Committee (IACUC) at Fu Jen Catholic University (approval number A10739). In this study, all efforts were made to minimize the number of animals used and their suffering.

### 2.3. Synaptosomes

Synaptosomes were isolated from the rat cerebral cortex as described [26,27]. Briefly, the cerebral cortex was homogenized in a medium containing 0.32 M sucrose (pH 7.4). The homogenate was centrifuged at 3000× *g* for 2 min at 4 °C and the supernatant spun again at 21,200× *g* for 10 min. The pellet was resuspended and gently layered on a Percoll discontinuous gradient and centrifugated at 32,500× *g* for 7 min. The layer (synaptosomal fraction) between 10 and 23% Percoll was collected, diluted in HEPES buffer medium (10 mM HEPES, 140 mM NaCl, 5 mM KCl, 1 mM MgCl_2_·6H_2_O, 5 mM NaHCO_3_, 1.2 mM Na_2_HPO_4_, and 10 mM glucose; pH 7.4). After centrifugation (27,000× *g* for 10 min), the pellet was resuspended in HEPES buffer medium, and the protein content was determined by the Bradford assay. Aliquots of 0.5 mg of synaptosomal suspension were centrifugated at 3000× *g* for 10 min to obtain the final synaptosome pellet.

### 2.4. Glutamate Release

Glutamate release assay was performed using enzyme-linked fluorescent detection of the released glutamate [24]. Synaptosomes (0.5 mg/mL) were resuspended in the HEPES buffer medium and placed in a LS-55 spectrofluorimeter (PerkinElmer Life Sciences, Waltham, MA, USA) at 37 °C with stirring. Nicotinamide adenine dinucleotide phosphate (NADP^+^, 1 mM), glutamate dehydrogenase (50 units/mL), and CaCl_2_ (1.2 mM) were added. After 3 min incubation, 4-aminopyridine (1 mM) was added to stimulate glutamate release. The fluorescence of NADPH was measured at 340 nm excitation and 460 nm emission and calibrated by glutamate standard (5 nmol). The released glutamate was expressed as nanomoles of glutamate per milligram of synaptosomal protein (nmol/mg). Release values quoted in the text represent levels of glutamate cumulatively released after 5 min depolarization and are indicated as nmol/mg/5 min.

### 2.5. Intrasynaptosomal Ca^2+^ Concentration

Intraterminal Ca^2+^ concentration ([Ca^2+^]_i_) was monitored in synaptosomes using Fura-2 AM in real time using a spectrofluorimeter according to a previous method [24]. Briefly, synaptosomes were resuspended and incubated in HEPES buffer medium containing Fura 2-AM (5 μM) and CaCl_2_ (0.1 mM) for 30 min at 37 °C in darkness. The suspension was centrifuged for 1 min at 10,000× *g*. Pellets were resuspended in HEPES buffer medium, transferred to a quartz cuvette, and measured with a LS-55 spectrofluorimeter at the excitation wavelengths of 340 and 380 nm (emission wavelength, 505 nm). Data were calculated through calibration procedures and equations described previously [28].

### 2.6. Synaptosomal Plasma Membrane Potential

Membrane potential was determined using the potentiometric fluorescent dye DiSC_3_(5) (5 µM) based on its potential-mediated binding to the plasma membrane, which was reported previously [24]. Data were gathered at 2 s intervals, and results are expressed in fluorescence units.

### 2.7. Animal Procedures and Histological Analyses

Rats were divided into four groups as follows: (i) dimethylsulfoxide (DMSO)-treated group (control); (ii) KA-treated group; (iii) silymarin 50 mg/kg + KA group; and (iv) silymarin 100 mg/kg + KA group. Silymarin (50 and 100 mg/kg) was dissolved in a saline solution containing 1% DMSO and was intraperitoneally (IP) administered 30 min before KA (15 mg/kg in 0.9% NaCl, pH 7.0, IP) injection. Following KA injection, rats were monitored for seizure behavior for 3 h [29]. Three days after KA treatment, rats were anesthetized with chloral hydrate (650 mg/kg, IP) and perfused for 7 min with normal heparinized saline and for 30 min with 4% paraformaldehyde in 0.1 M phosphate-buffered saline (PBS). Brains were then removed and postfixed overnight in 4% paraformaldehyde at 4 °C, then immersed in 30% sucrose phosphate buffer for 24 h. Serial coronal sections of 30 μm thickness were cut on a freezing microtome. For neutral red staining, the sections were mounted on gelatin-coated slides and stained with 1% neutral red. Each stained section was observed with a light microscope to assess the degree of neuronal loss within the hippocampus [30].

Immunofluorescence staining was performed on another set of series, as previously described [29]. Briefly, the sections were washed with PBS and blocked with 2% normal goat serum containing 0.3% Triton X-100 in PBS for 1 h at room temperature. Sections were then incubated overnight at 4 °C with appropriate primary antibodies (anti-NeuN, 1:500, Abcam; anti-OX42, 1:2000, Merck Milipore; anti-GFAP, 1:2000, Cell Signaling, Danvers, MA, USA). The sections were then incubated with IgG-DyLight 594 (1:1000, Vector Laboratories, Burlingame, CA, USA) and FITC-conjugated IgG (1:1000, Vector Laboratories) for 1 h. Nuclei were counterstained in DAPI (1 μg/mL, Sigma-Aldrich) for 20 s. After washing with PBS, sections were mounted onto gelatin-coated glass slides and coverslipped with Entellan mounting medium (Merck). Images were acquired in an upright fluorescence microscope (Zeiss Axioskop 40, Goettingen, Germany) and Image Xpress micro confocal (Molecular Devices, San Jose, CA, USA). The number of living neurons or NeuN+, OX42+, and GFAP+ cells was measured in an area of 255 × 255 μm of the hippocampus using computer-assisted image analysis system (Image J; NIH Image, National Institutes of Health, Bethesda, MD, USA).

### 2.8. Determination of Glutamate by High-Performance Liquid Chromatography

The frozen hippocampus tissues were homogenized in 500 μL of 200 mM ice-cold perchloric acid and centrifuged at 1500× *g* for 10 min at 4 °C. The supernatant (10 μL) was filtered and injected directly into a high-performance liquid chromatography system with electrochemical detection (HTEC-500). Peak heights were recorded with an integrator, and the concentration of glutamate was calculated on the basis of known standards [29].

### 2.9. Western Blot

Synaptosomes or the frozen hippocampus tissues were suspended in ice-cold lysis buffer and quantified for protein content [24]. Thirty micrograms of protein was subjected to 8–12% SDS-PAGE, and proteins were electroblotted to nitrocellulose membranes. After blocking with 5% nonfat milk for 1 h, membranes were incubated overnight at 4 °C with the following antibodies: anti-phospho-ERK1/2 (p-ERK1/2,1:2000, Cell Signaling), anti-ERK1/2 (1:4000, Cell Signaling), anti-HSP70 (1:2000, Enzo), anti-Akt (1:6000, Cell Signaling), anti-phospho-Akt (p-Akt,1:6000, Cell Signaling), and anti-β-actin (1:8000, Cell Signaling). Membranes were then washed three times with Tris-buffered saline and incubated for 1 h at room temperature with suitable secondary antibodies. After washing, the blots were developed using an enhanced chemiluminescence detection system (Amersham, Berkshire, UK), which was followed by analysis using the Image J software (version 1.45 J, National Institutes of Health, Bethesda, Rockville, MD, USA).

### 2.10. Statistical Analyses

Statistical analyses were performed using SPSS (v.24; IBM, Armonk, NY, USA). Data are presented as mean ± standard error of the mean (SEM). One-way analysis of variance test (ANOVA) with Tukey’s post hoc test was used to analyze three or more group data. For two-group comparison, Student’s unpaired *t*-test was used to determine statistical differences between groups. For all analyses, *p* < 0.05 was considered to be statistically significant.

## 3. Results

### 3.1. Silymarin Decreases 4-Aminopyridine-Evoked Glutamate Release from Rat Cerebrocortical Synaptosomes by Reducing Exocytotic Mechanism

As can be seen in Figure 1A, the exposure of cerebrocortical synaptosomes to a mild depolarizing stimulus (1 mM 4-aminopyridine) elicited a significant release of glutamate in the presence of 1.2 mM CaCl_2_. Preincubation with silymarin (30 µM) for 10 min before 4-aminopyridine addition had no influence on basal glutamate release, but it caused inhibition of 4-aminopyridine-evoked release of about 58% (*t*(10) = 25.6, *p* < 0.001). The effect of silymarin was concentration-dependent with an IC_50_ of about 12µM; the maximal inhibition of evoked glutamate release was observed at 30 µM. Silymarin did not cause a complete blockade of release even at the highest concentrations used (50 µM) (Figure 1B). Given the robust depression of glutamate release that was seen with 30 µM silymarin, this concentration of silymarin was used in subsequent experiments to investigate the mechanisms that underlie the ability of silymarin to reduce glutamate release. In Figure 1C, we investigated whether the effect of silymarin on 4-aminopyridine-evoked glutamate release was mediated by an effect on exocytosis (Ca^2+^-dependent release) or reversal of the glutamate transporter (Ca^2+^-independent release) [26]. First, Ca^2+^-independent glutamate release was measured in a extracellular Ca^2+^-free solution that contained 300 µM EGTA. Under these conditions, 4-aminopyridine-evoked glutamate release was significantly reduced (*p* < 0.001). This Ca^2+^-independent release evoked by 4-aminopyridine was, however, not affected by silymarin (30 µM) (*F*(2.13) = 177.2, *p* = 1). Second, the vesicular glutamate transporter inhibitor bafilomycin A1 (0.1 µM) reduced 4-aminopyridine-evoked glutamate release (*p* < 0.001) and completely prevented the action of silymarin (30 µM). In the five tested synaptosomal preparations, no statistical difference was observed between the release after bafilomycin A1 alone and after bafilomycin A1 and silymarin (*F*(2.12) = 299.7, *p* = 0.9; Figure 1C). Additionally, the effect of silymarin on [Ca^2+^]_i_ is shown in Figure 1D. 4-aminopyridine (1 mM) caused an increase in [Ca^2+^]_i_ to a plateau level, which was decreased by 70% when synaptosomes were pretreated with silymarin (30 µM) (*t*(8) = 18.4, *p* < 0.001). At the concentration applied, silymarin did not significantly affect basal [Ca^2+^]_i_, (*p* = 0.9).

### 3.2. Reduced Ca^2+^ Influx through N- and P/Q-Type Ca^2+^ Channels Is Involved in the Silymarin Effect

Figure 2 shows that 4 μM ω-CgTX MVIIC, a N- and P/Q-type Ca^2+^ channel inhibitor, 10 μM dantrolene, an inhibitor of ryanodine receptors, and 10 μM CGP37157, an inhibitor of mitochondrial Na^+^/Ca^2+^ exchanger, significantly inhibited glutamate release evoked by 4-aminopyridine (*p* < 0.001), consistent with the involvement of voltage-dependent Ca^2+^ channels, ryanodine receptors, and mitochondrial Na^+^/Ca^2+^ exchanger in the releasing effect [31,32]. Silymarin (30 µM) significantly decreased 4-aminopyridine-evokted glutamate release (*p* < 0.001). With ω-CgTX MVIIC present, however, silymarin (30 µM) failed to further reduce 4-aminopyridine-evokted glutamate release. The release measured in the presence of ω-CgTX MVIIC and silymarin was similar to that obtained in the presence of ω-CgTX MVIIC alone (*F*(2.12) = 434.8, *p* = 1). On the contrary, silymarin (30 µM) still effectively inhibited 4-aminopyridine-evoked glutamate release in the presence of dantrolene (*F*(2.13) = 168.4, *p* < 0.001) or CGP37157 (*F*(2.12) = 211.5, *p* < 0.001). The figure also shows that 1 µM xestospongin C, a membrane-permeant IP3 receptor antagonist, failed to affect glutamate release caused by 4-aminopyridine (*p* = 0.7), suggesting that the IP3 receptor was not involved. With xestospongin C present, silymarin (30 µM) significantly inhibited 4-aminopyridine-evoked glutamate release (*F*(2.12) = 102.1, *p* < 0.001; Figure 2). The lack of additivity in the inhibitory actions of silymarin and ω-CgTX MVIIC on glutamate release suggested that there was a preferential interaction between the pathway mediated by silymarin and N- and P/Q-type Ca^2+^ channels. In contrast, the additivity in the actions of silymarin and dantrolene or CGP37157 on glutamate release indicated that the action of silymarin was not due to a reduction of Ca^2+^ release from intracellular stores.

### 3.3. Silymarin Does Not Affect Synaptosomal Membrane Potential

The effect of silymarin on synaptosomal membrane potential using the membrane potential-sensitive dye DiSC_3_(5) is shown in Figure 3A. 4-aminopyridine (1 mM) caused a significant increase of DiSC_3_(5) fluorescence in the synaptosomes. Application of silymarin (30 µM) had no effect on either the basal DiSC_3_(5) fluorescence or 4-aminopyridine-evoked increase in DiSC_3_(5) fluorescence (*t*(8) = −0.14, *p* = 0.9). Figure 3B shows that the exposure of synaptosomes to 15 mM KCl elicited glutamate release, which involves only Ca^2+^ channel activation [33]. Addition of silymarin (30 µM) also inhibited KCl (15 mM)-evoked release by 54% (*t*(8) = 61.6; *p* < 0.001). In addition, Figure 3C shows that the Na^+^ channel activator veratridine (50 µM) triggered glutamate release, which was unaffected by silymarin (30 µM) (*t*(8) = 0.83; *p* = 0.4).

### 3.4. Suppressed MAPK Pathway Is Involved in Silymarin-Mediated Inhibition of Glutamate Release

As shown in Figure 4, we used H89, a PKA inhibitor, GF109203X, a PKC inhibitor, and PD98059, a MAPK inhibitor, to elucidate the intraterminal enzymatic pathways involved in silymarin-mediated inhibition of glutamate release. The control 4-aminopyridine (1 mM)-evoked glutamate release was decreased by silymarin (30 µM) (*p* < 0.001). With H89 (100 µM) or GF109203X (10 µM) present, silymarin (30 µM) still effectively inhibited 4-aminopyridine-evoked glutamate release. A statistical difference was observed between the release after H89 or GF109203X alone and after H89 or GF109203X and silymarin treatment (H89 + silymarin, *F*(2,12) = 343.4, *p* < 0.001; GF109203X + silymarin, *F*(2,12) = 410.2, *p* < 0.001). In contrast, the inhibitory effect of silymarin (30 µM) on 4-aminopyridine-stimulated glutamate release was not observed in the presence of PD98059 (50 µM). No statistical difference was observed between the release after PD98059 alone and after PD98059 and silymarin (*F*(2,12) = 40.8, *p* = 0.5). A similar result was also obtained using FR180204 (10 µM), a potent ERK1/2 inhibitor. The release measured in the presence of FR180204 and silymarin was similar to that obtained in the presence of FR180204 alone (*F*(2,13) = 236.1, *p* = 0.5). At the concentration applied, all inhibitors significantly reduced the release of glutamate evoked by 4-aminopyridine (1 mM) (*p* < 0.001). In addition, as shown in Figure 5, 4-aminopyridine (1 mM) increased the phosphorylation of ERK1/2 (*p* < 0.001). Silymarin at a concentration effective in inhibiting glutamate release also significantly decreased 4-aminopyridine-induced ERK1/2 phosphorylation (*F*(2,15) = 9.7, *p* < 0.001).

### 3.5. Silymarin Attenuats Seizures and Glutamate Concentration Elevation in KA-Treated Rats

The experimental design is shown in Figure 6A. We further examined whether silymarin executed a protective action in a rat model of excitotoxicity induced by KA. As can be seen in Figure 6, silymarin at the doses of 50 and 100 mg/kg administrated (IP) 30 min prior to KA (15 mg/kg, IP) delayed the latent period to start seizure (*F*(2,20) = 139.5, *p* < 0.001; Figure 6B) and decreased the severity of the seizure in comparison with the KA group (*F*(2,20) = 28.1, *p* < 0.001; Figure 6C). In addition, in agreement with previously published data [29,30], a significant elevation of glutamate levels in the hippocampus was observed at 72 h after KA administration (IP) (*p* < 0.001). The group pretreated with silymarin (10 or 50 mg/kg, IP) offered significant restoration of glutamate levels in comparison to the KA-only group (*F*(3,8) = 57.1, *p* < 0.001; Figure 6D).

### 3.6. Silymarin Attenuats Hippocampal Neuronal Damage in KA-Treated Rats

Neuronal damage in a rat model of excitotoxicity induced by KA was assessed by neutral red and NeuN staining. Seventy-two hours after administration of KA (15 mg/kg, IP), neutral red staining clearly showed a significant neuronal loss in the CA1 and CA3 regions of the hippocampus compared to the DMSO-treated rats (control) (*p* < 0.001). Pretreatment with silymarin (50 or 100 mg/kg, IP) 30 min before KA administration increased the neuronal survival rate compared to KA treatment alone (CA1, *F*(3,11) = 169.1, *p* < 0.001; CA3, *F*(3,11) = 19.6, *p* < 0.001; Figure 7A,B). A similar preventive effect of silymarin against neuronal death was also observed with the NeuN neuronal marker. As shown in Figure 8A, we observed a decrease in immunoreactivity for NeuN mainly in the CA1 and CA3 regions of KA-treated rats compared to the DMSO-treated rats (control) (*p* < 0.001). In comparison to the KA group, NeuN immunoreactivity was higher in the CA1 and CA3 regions of rats pretreated with silymarin. Statistical analysis by one-way ANOVA corroborated these findings, showing that KA administration resulted in a significant decrease in the number of NeuN+ cells, and that silymarin pretreatment significantly suppressed this effect (CA1, *F*(3,8) = 71.5, *p* < 0.001; CA3, *F*(3,8) = 143.4, *p* < 0.001; Figure 8B).

### 3.7. Silymarin Attenuates Microglia and Astrocyte Activations in KA-Treated Hippocampus

The activation of microglia was determined with immunostaining of a microglial marker OX42. As shown in Figure 9A, microglial cells in the hippocampus of the control group exhibited a resting morphology with small cell bodies and thin processes. Conversely, microglial cells in the hippocampus of KA-injected rats displayed enlarged cell bodies with considerably shorter and thicker processes (indicating the activation state). In rats pretreated with silymarin, most of the microglial cells were in a ramified or resting state. One-way ANOVA revealed a significant increase of OX42+ cells in the hippocampus of the KA group (*p* < 0.001) compared to the control group, whereas the silymarin groups did not differ from the control group (*p* > 0.05). However, a significant decrease in the number of OX42+ cells was observed in the hippocampus of the silymarin groups compared to the KA group (CA1, *F*(3,8) = 123.4, *p* < 0.001; CA3, *F*(3,8) = 102.4, *p* < 0.001; Figure 9B). As was observed for microglial cells, activated astrocytes (which immunostained positively for GFAP) were rarely observed in the control group (Figure 10A). However, activated astrocyte proportions were significantly increased in the hippocampus after KA injection, and silymarin pretreatment significantly reduced these proportions (CA1, *F*(3,8) = 256.6, *p* < 0.001; CA3, *F*(3,8) = 236.7, *p* < 0.001; Figure 10B).

### 3.8. Silymarin Inhibits HSP70 Expression and Increases Akt Phosphorylation in the Hippocampus of KA-Treated Rats

Figure 11A shows the effect of silymarin on the expression of heat shock protein 70 (HSP70), a cellular stress marker that is involved in KA-induced excitotoxicity [34,35]. Using western blot analysis, the expression level of HSP70 in the hippocampus of KA-group was significantly higher than that of the control group (*p* < 0.001). In rats pretreated with silymarin, the HSP70 expression induced by KA was decreased compared to the KA group (*F*(3,8) = 189.7, *p* < 0.001). Figure 11B shows the effect of silymarin on the phosphorylation of protein kinase B (Akt), which often leads to cell survival. In contrast to HSP70 expression, phosphorylated Akt decreased significantly in the hippocampus of the KA-treated group compared to the control group. Akt phosphorylation levels were significantly upregulated in groups pretreated with silymarin compared to the KA group (*F*(3,8) = 4.5, *p* < 0.05).

## 4. Discussion

Excess glutamate causes neuronal injury and may lead to neurological diseases [5]. Excessive glutamate release can be inhibited by natural compounds present in foods and medicinal plants [30,36,37]. Therefore, the present work investigated the modulation of the milk thistle polyphenoic flavonoid silymarin on glutamate release from rat cerebrocortical synaptosomes and evaluated whether silymarin exerts a neuroprotective effect in an in vivo rat model of KA-induced excitotoxicity.

We found that silymarin at doses unable to affect basal glutamate release substantially reduced 4-aminopyridine-induced glutamate release from rat cerebrocortical nerve terminals. Because inhibition was observed during 4-aminopyridine stimulation, it could have resulted from (i) inhibition of Ca^2+^-dependent vesicular release or (ii) Ca^2+^-independent nonvesicular release through glutamate transporter reversal [38,39]. In the present study, silymarin inhibited 4-aminopyridine-stimulated glutamate release in the presence of extracellular Ca^2+^ but not in the absence of extracellular Ca^2+^, indicating that silymarin reduces glutamate release by inhibiting voltage-dependent Ca^2+^ channels. This inference is further supported by the following observations. First, the vesicular transporter inhibitor bafilomycin A1 completely prevented the effect of silymarin on 4-aminopyridine-induced glutamate release. Second, silymarin significantly reduced 4-aminopyridine-induced increase in [Ca^2+^]_i_. Third, the inhibition of N- and P/Q-type Ca^2+^ channels abolished silymarin’s effect, suggesting that silymarin inhibits glutamate release by decreasing extracellular Ca^2+^ entry through N- and P/Q-type Ca^2+^ channels that trigger glutamate release from synaptosomes [32,40]. Finally, the action of silymarin was totally insensitive to intracellular Ca^2+^ release inhibitors—dantrolene, CGP37157, or xestospongin C—thereby excluding the involvement of intracellular Ca^2+^ stores [31]. Our data imply that silymarin inhibits glutamate release by suppressing N- and P/Q-type Ca^2+^ channel activity. However, whether a direct interaction exists between silymarin and presynaptic voltage-dependent Ca^2+^ channels remains to be explored.

Silymarin was also found to inhibit the release of glutamate evoked by moderate K^+^ depolarization (15 mM) of rat cerebrocortical nerve terminals, which excludes the involvement of Na^+^ channels. This is because 4-aminopyridine-induced glutamate release involves Na^+^ and Ca^2+^ channels, whereas 15 mM KCl-evoked glutamate release involves only Ca^2+^ channels [33,41]. Our inference is further supported by the observation that silymarin did not affect veratridine-induced glutamate release. Additionally, silymarin did not affect 4-aminopyridine-evoked synaptosomal membrane potential depolarization measured with a dye of DiSC_3_(5), suggesting the lack of involvement of K^+^ channel blockade. Taken together, our data indicate that silymarin-mediated inhibition of glutamate release is not due to Na^+^ channel inhibition or K^+^ channel activation, leading to a subsequent decrease in voltage-dependent Ca^2+^ influx.

The silymarin-mediated inhibition of 4-aminopyridine-induced glutamate release was largely abolished by the inhibition of MAPK or ERK1/2 but was unaffected by the inhibition of PKA or PKC. On the basis of this finding, together with the inhibition of 4-aminopyridine-induced ERK1/2 phosphorylation by silymarin, we suggest that silymarin inhibits glutamate release by suppressing terminal ERK1/2. Phosphorylative processes mediated by ERK1/2 affect glutamate exocytosis from rat cortical glutamatergic synaptosomes [42,43]. In nerve terminals, increased [Ca^2+^]_i_ activates ERK1/2, which phosphorylates several synaptic proteins, especially synapsin I. This phosphorylation process promotes the dissociation of synaptic vesicles from the actin cytoskeleton and increases the number of available synaptic vesicles, leading to increased glutamate release [44,45,46]. Thus, silymarin-mediated inhibition of N- and P/Q-type Ca^2+^ channels may suppress ERK1/2 activation, leading to reduced exocytotic glutamate release.

Numerous studies have highlighted the neuroprotective effects of silymarin [21,47], but to our knowledge, no study has evaluated it in an animal model of KA-induced excitotoxicity. The synaptosomal observations led us to propose that silymarin contributes to neuronal survival by inhibiting glutamate release and limiting excitotoxicity. Supporting this hypothesis, we found that silymarin exerted a preventive effect against neurological damage in a rat model of excitotoxicity induced by KA administration. KA is commonly used to induce excitotoxicity because it is a glutamate analog that activates glutamate receptors and leads to excessive glutamate release and seizures [22,48]. Similarly, in the current study, KA administration (15 mg/kg, IP) caused seizures and increased glutamate levels in the hippocampus. However, pretreatment with silymarin (50 or 100 mg/kg, IP) 30 min before KA could reverse these KA-induced trends. In addition, systemic administration of KA has been shown to cause neuronal death in many regions of the brain, especially in the CA1 and CA3 regions of the hippocampus [49,50,51], and this is in consistent with our results, as evaluated using neutral red and NeuN staining. Our data demonstrated that silymarin pretreatment in the presence of KA substantially attenuated neuronal damage in the CA1 and CA3 areas. Additionally, microglia and astrocyte activation in the CA1 and CA3 regions was observed after KA administration in this study, which is consistent with previous reports [52,53]. The present study also showed the anti-inflammatory property of silymarin as our results demonstrated lower degrees of activated microglia and astrocytes in rats pretreated with silymarin in the presence of KA. Silymarin could possibly prevent hippocampal neuronal damage by suppressing inflammatory processes in KA-treated rats. However, how silymarin affects glia activation remains unclear. Glia activation and the consequent production of proinflammatory cytokines are believed to contribute to KA-induced neuronal damage [52,54]. Thus, the possible involvement of suppression of proinflammatory cytokine production in the anti-inflammatory effect of silymarin observed in the present study should be considered. As already observed in previous studies, silymarin exerts neuroprotection against 6-hydroxydopamine- or aluminum chloride-induced neurological deficits by means of suppressing glial cell activation and inflammatory cytokine (interleukin-1β, interleukin-6, and tumor necrosis factor-α) production [55,56].

In the hippocampus of KA-treated rats, we found a significant increase in HSP70 expression, which was suppressed by silymarin pretreatment. HSP70 is an inducible heat-shock protein, and its expression is a marker of cellular stress [57]. Higher HSP70 levels that accompany neuronal damage have been detected in the brains of KA animal models, and KA-induced excitotoxic injury also appears to be prevented by HSP70 inhibition [34,35,58]. Although the role of HSP70 in excitotoxic insults is controversial [59,60], our results indicate that silymarin prevents hippocampal neurons against KA-induced cellular insult by inhibiting HSP70 expression. Furthermore, we have demonstrated that silymarin can activate the prosurvival Akt signaling cascade in the KA-induced rat excitotoxicity model. Akt activation may, in part, increase the ability of neuronal cells to survive toxic insults [61,62]. In this study, KA reduced hippocampal p-Akt levels, which is consistent with previous reports [63,64,65]. In the presence of silymarin, p-Akt levels remained high in the hippocampus. This supports the idea of a causal relationship between neuroprotection observed with silymarin treatment and Akt pathway activation in the hippocampus. Although exact mechanisms underlying the neuroprotective effects of silymarin remains unclear, several mechanisms, such as antioxidation, free radical scavenging, and suppression of inflammation have been proposed [47,66,67]. In addition, our data suggest that the antiexcitotoxic effect of silymarin may partly contribute to its neuroprotective effects on the CNS.

Silymarin is a promising therapeutic candidate for CNS diseases not only because of its neuroprotective effects but also its safety. Silymarin was safe up to 4500 mg/kg orally when tested in rodents after three months of exposure [10]. Moreover, oral administration of therapeutic or high doses of silymarin had no toxicity or adverse effects in healthy and patient volunteers [68,69]. The dose of silymarin used to prevent neuronal damage in our in vivo study was 50–100 mg/kg. Consistent with our study, numerous studies have revealed that the effect of silymarin occurs at a high dose (200–300 mg/kg) in vivo [20,70,71].

## 5. Conclusions

In conclusion, silymarin inhibited glutamate release in rat cortical nerve endings and exerted a preventive effect against neurological damage on rats with KA-induced excitotoxicity (Figure 12). Our data suggest that silymarin can be a potential agent for treating diseases related to neuronal excitotoxicity due to excessive release and accumulation of glutamate. Additional studies are required to establish the validity of these findings in humans.

## Figures and Tables

**Figure 1 biomedicines-08-00486-f001:**
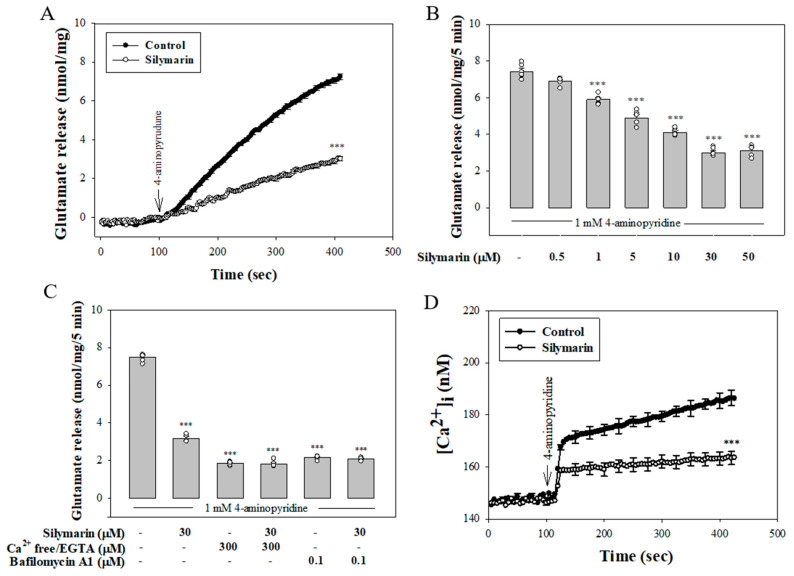
Silymarin inhibits 4-aminopyridine-evoked glutamate release and intracellular Ca^2+^ elevation in rat cerebral cortex synaptosomes. (**A**) Glutamate release was evoked by the addition of 1 mM 4-aminopyridine in the absence (control) and presence of 30 µM silymarin added 10 min before depolarization. (**B**) Concentration-dependent inhibition of 4-aminopyridine-evoked glutamate release by silymarin. (**C**) Effect of silymarin on 4-aminopyridine-evoked glutamate release in extracellular Ca^2+^-free solution or in the presence of the vesicular transporter inhibitor bafilomycin A1. (**D**) Silymarin decreases 4-aminopyridine-evoked increase in [Ca^2+^]_i_. Data are mean ± standard error of the mean (SEM) (*n* = 5–6 per group). *** *p* < 0.001 (in comparison with the control).

**Figure 2 biomedicines-08-00486-f002:**
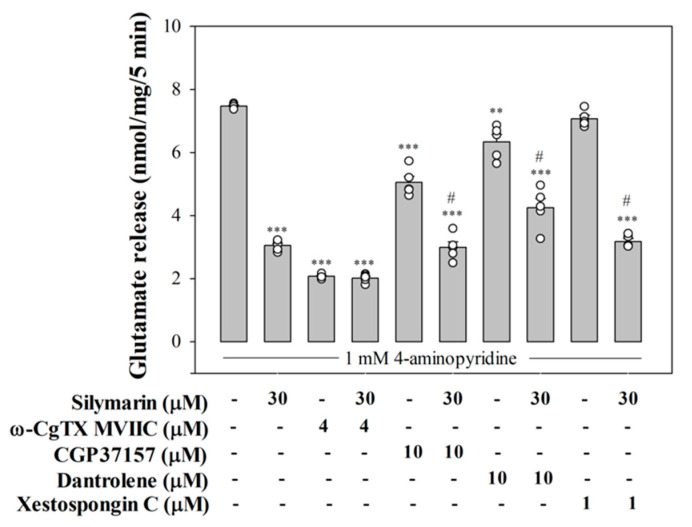
Silymarin-mediated inhibition of 4-aminopyridine-evoked glutamate release in the presence of N- and P/Q-type Ca^2+^ channel blocker ω-CgTX MVIIC, ryanodine receptor inhibitor dantrolene, mitochondrial Na^+^/Ca^2+^ exchanger inhibitor CGP37157, or IP3 receptor antagonist xestospongin C. Silymarin was added 10 min before the addition of 4-aminopyridine, and other drugs were added 10 min before this. Data are mean ± SEM (*n* = 5 per group). ** *p* < 0.01, *** *p* < 0.001 (in comparison with the control), # *p* < 0.001 (compared with the dantrolene-, CGP37157-, or xestospongin C-treated group).

**Figure 3 biomedicines-08-00486-f003:**
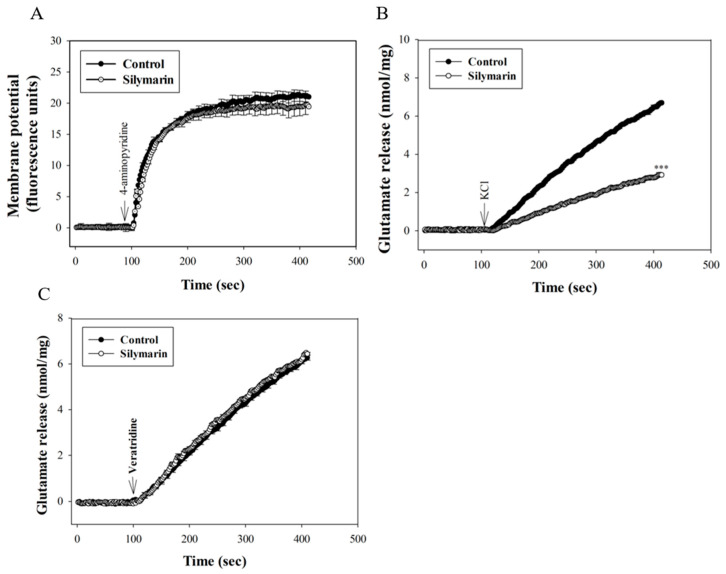
Effect of silymarin on the synaptosomal membrane potential (**A**) and the release of glutamate evoked by 15 mM KCl (**B**) or 50 µM veratridine (**C**). Silymarin (30 µM) was added 10 min before the addition of 4-aminopyridine or KCl. Data are mean ± SEM (*n* = 5 per group). *** *p* < 0.001 (in comparison with the control).

**Figure 4 biomedicines-08-00486-f004:**
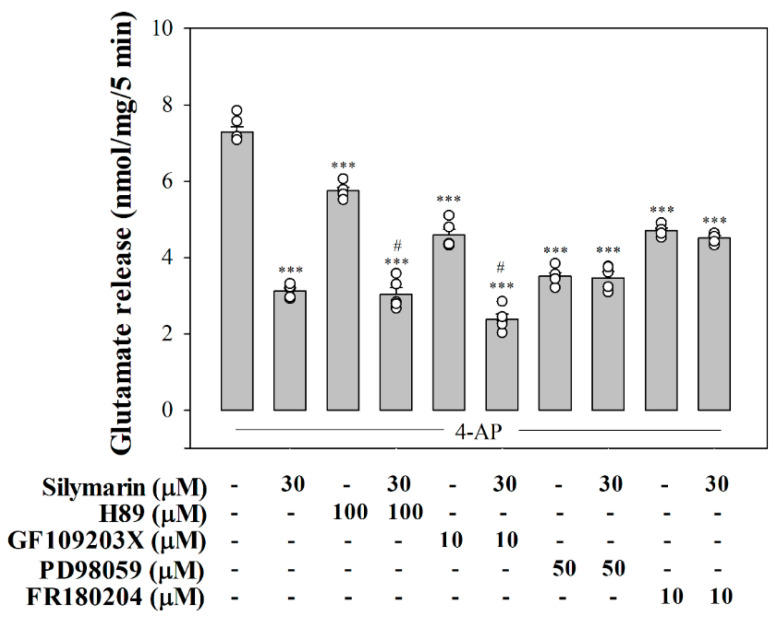
Effect of the protein kinase A (PKA) inhibitor H89, the protein kinase C (PKC) inhibitor GF109203X, the mitogen-activated protein kinase (MAPK) inhibitor PD98059, and the extracellular signal-regulated kinase 1/2 (ERK1/2) inhibitor FR180204 on silymarin-induced inhibition of evoked glutamate release. Silymarin was added 10 min before the addition of 4-aminopyridine, and other drugs were added 20 min before this. Data are mean ± SEM (*n* = 5–6 per group). *** *p* < 0.001 (in comparison with the control), # *p* < 0.001 (compared with the H89- or GF109203X-treated group).

**Figure 5 biomedicines-08-00486-f005:**
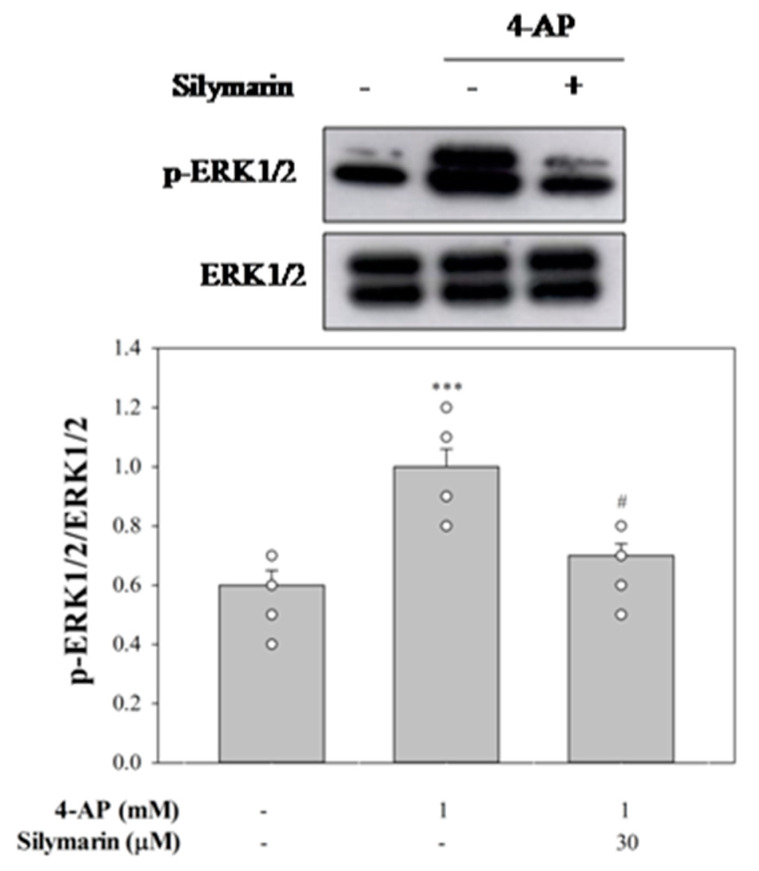
Effects of 4-aminopyridine-evoked depolarization and silymarin treatment on ERK1/2 phosphorylation. Silymarin was added 10 min before the addition of 4-aminopyridine. Data are mean ± SEM (*n* = 6 per group). *** *p* < 0.001 (in comparison with the control), # *p* < 0.001 (compared with the 4-AP-treated group).

**Figure 6 biomedicines-08-00486-f006:**
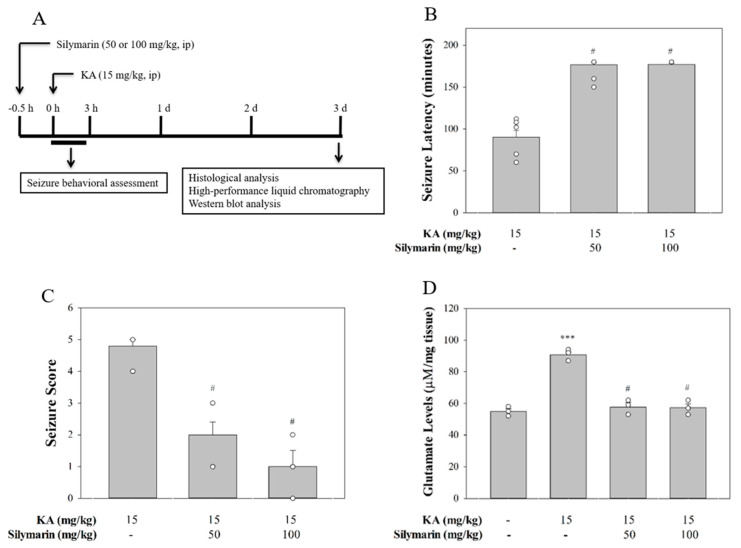
(**A**) Schematic timeline of experimental protocol. Seizure latency (**B**), seizure score (**C**), and glutamate levels (**D**) in the presence of silymarin versus animals that were injected only with kainic acid (KA). Data are mean ± SEM (*n* = 3–9 per group). *** *p* < 0.001 (in comparison with the control group), # *p* < 0.001 (compared with the KA group).

**Figure 7 biomedicines-08-00486-f007:**
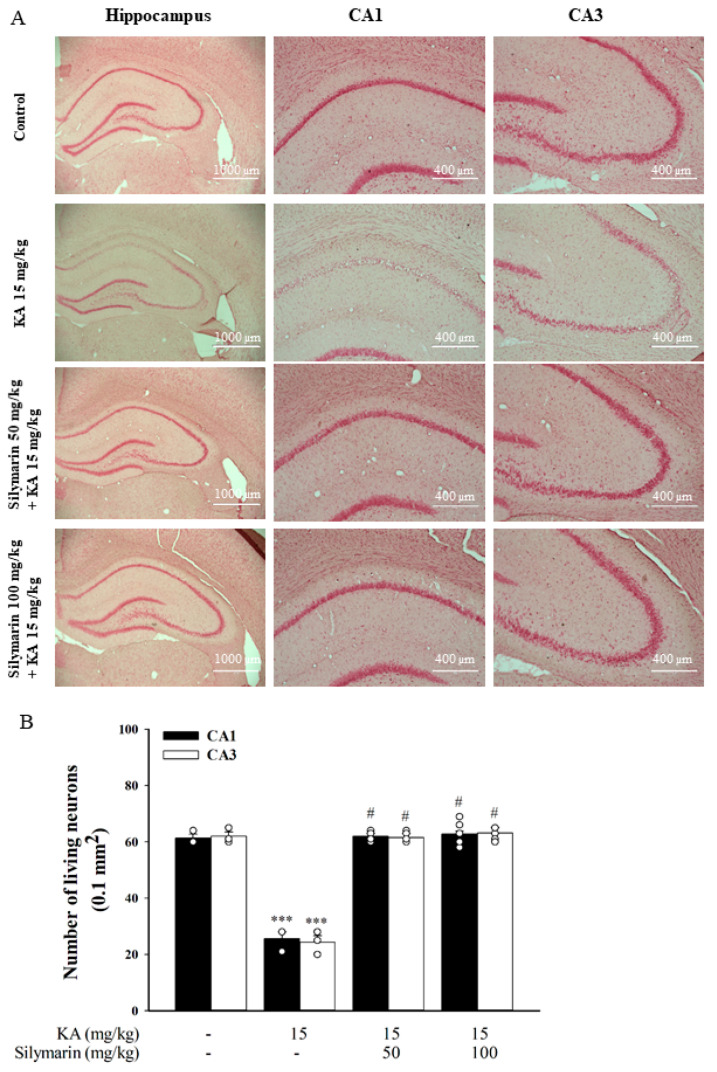
Effect of silymarin on KA-induced neuronal loss. Silymarin was intraperitoneally administrated (50 and 100 mg/kg, IP) 30 min before receiving KA injection (15 mg/kg, IP), and neuronal loss in the hippocampus were evaluated 72 h after KA treatment. (**A**) Representative images showing neutral red staining in the hippocampus. (**B**) The number of surviving neurons in the hippocampal CA1 and CA3 regions was counted. Data are mean ± SEM (*n* = 3–5 per group). *** *p* < 0.001 (in comparison with the control group), # *p* < 0.001 (compared with the KA group).

**Figure 8 biomedicines-08-00486-f008:**
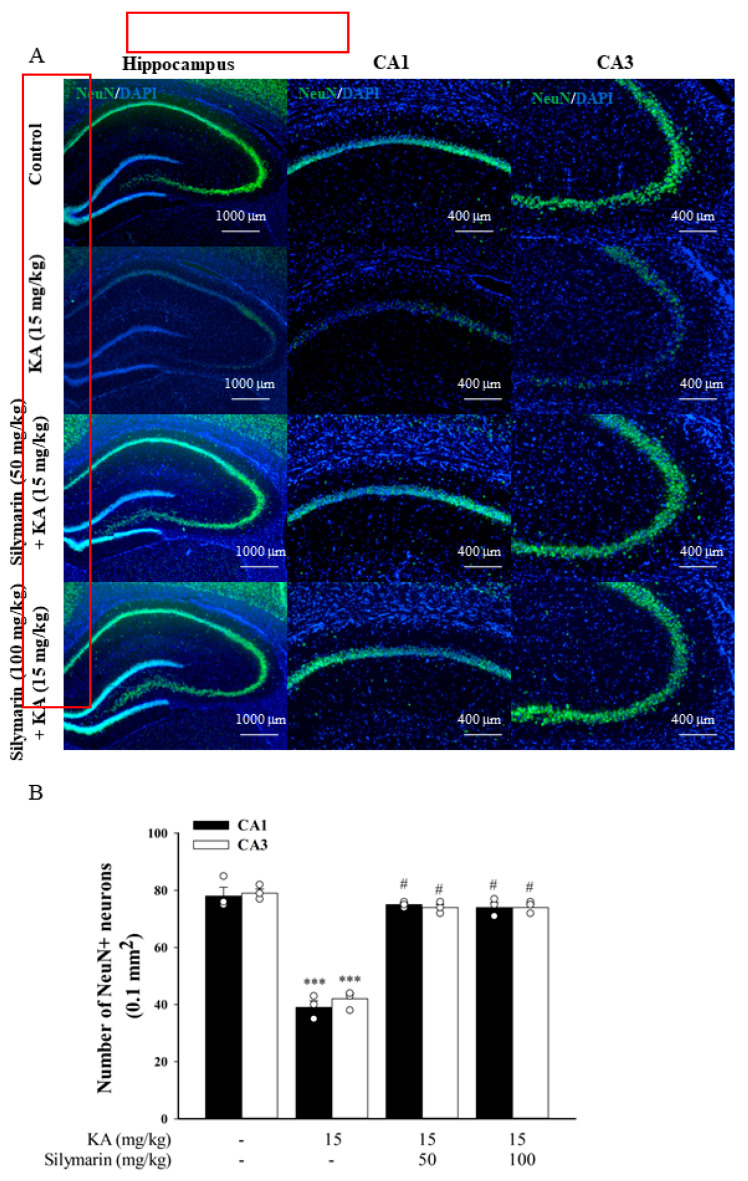
Representative (**A**) and quantitative (**B**) verification of prevention by silymarin on KA-induced neuronal cell death with NeuN. Data are mean ± SEM (*n* = 3 per group). *** *p* < 0.001 (in comparison with the control group), # *p* < 0.001 (compared with the KA group).

**Figure 9 biomedicines-08-00486-f009:**
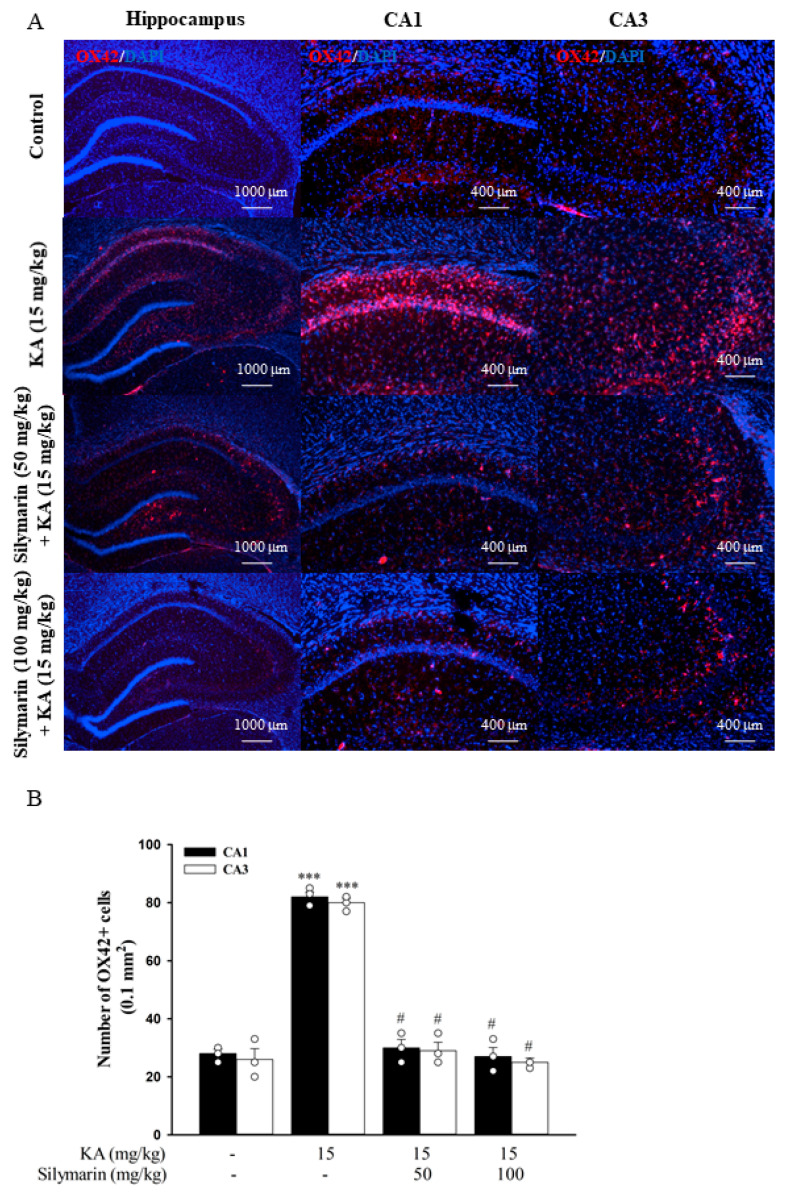
Photomicrographs of OX42 immunostaining (**A**) and their quantified immunopositive cells (**B**) in the hippocampal CA1 and CA3 regions. Data are mean ± SEM (*n* = 3 per group). *** *p* < 0.001 (in comparison with the control group), # *p* < 0.001 (compared with the KA group).

**Figure 10 biomedicines-08-00486-f010:**
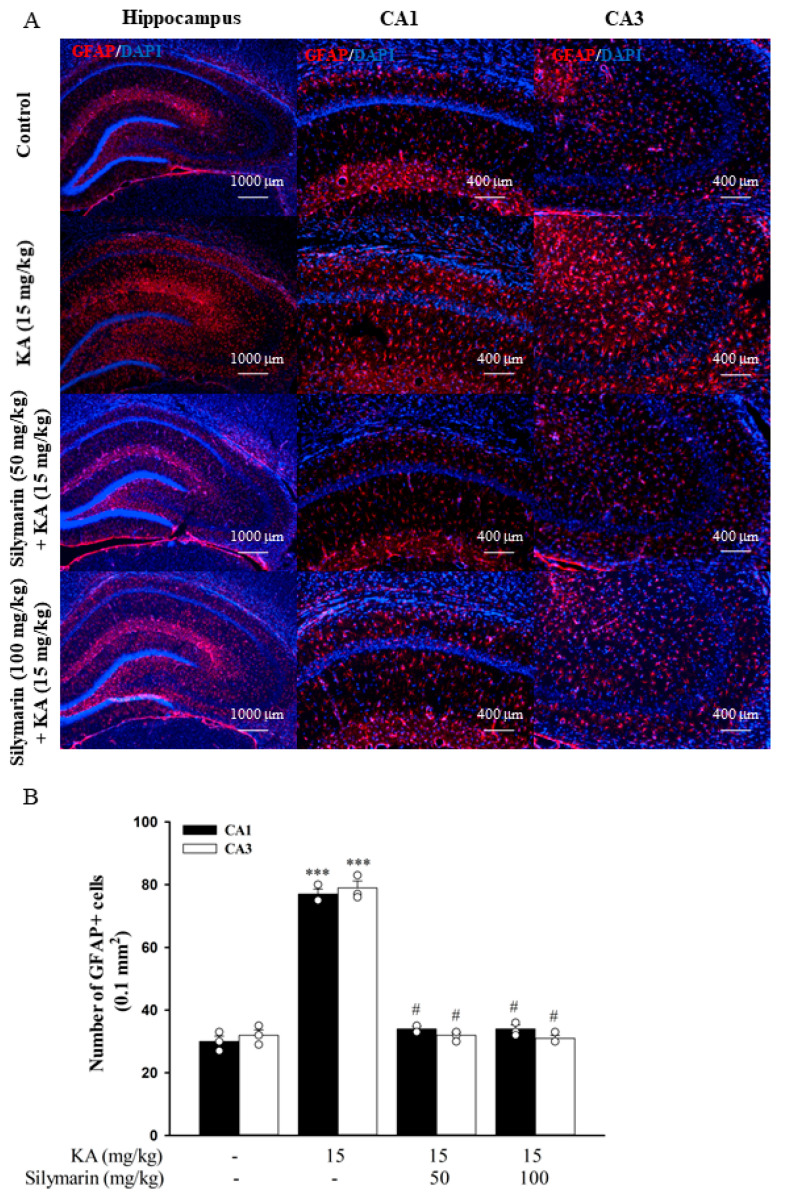
Photomicrographs of GFAP immunostaining (**A**) and their quantified immunopositive cells (**B**) in the hippocampal CA1 and CA3 regions. Data are mean ± SEM (*n* = 3 per group). *** *p* < 0.001 (in comparison with the control group), # *p* < 0.001 (compared with the KA group).

**Figure 11 biomedicines-08-00486-f011:**
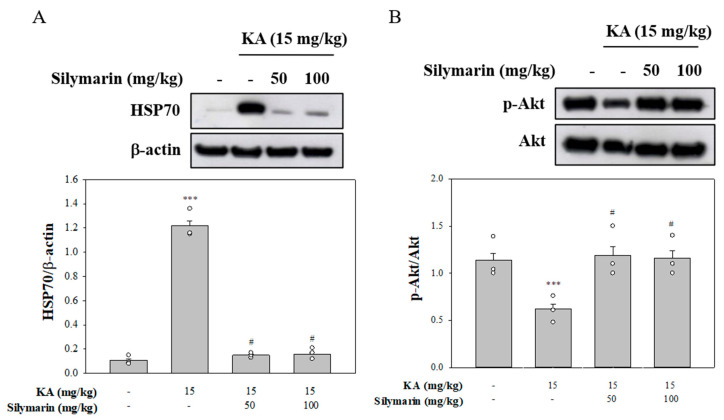
Effects of silymarin on the expression of heat shock protein 70 (HSP70) and p-Akt in KA-treated hippocampus. Western blots show hippocampal HSP70 (**A**) and p-Akt (**B**) expression 72 h after KA treatment. Data are mean ± SEM (*n* = 3 per group). *** *p* < 0.001 (in comparison with the control group), # *p* < 0.001 (compared with the KA group).

**Figure 12 biomedicines-08-00486-f012:**
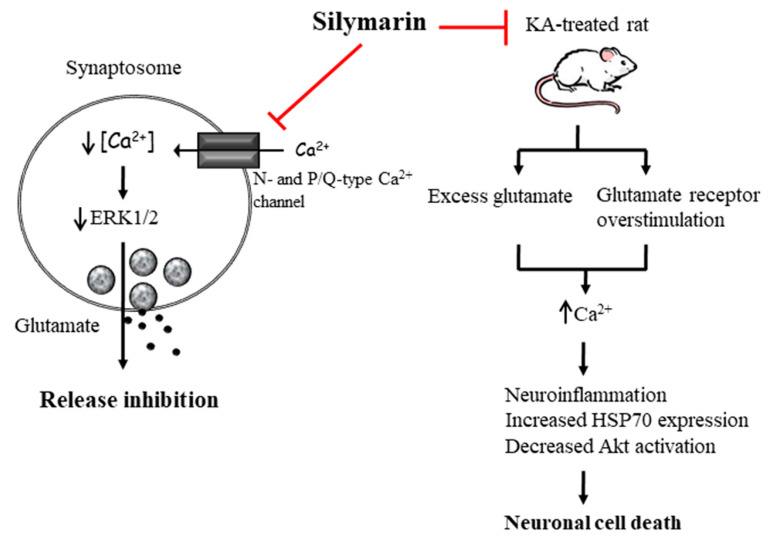
The proposed working model of silymarin depressing glutamate release from synaptosomes and having a preventive effect against KA-induced neuronal death. In rat synaptosomes, silymarin inhibits glutamate release by suppressing N- and P/Q-type Ca^2+^ channels and ERK1/2 activity. In a rat model of KA-induced excitotoxicity, silymarin pretreatment substantially attenuates KA-induced neuronal death by suppressing inflammatory processes and HSP70 expression and upregulating Akt activation. The red T-shaped lines represent inhibition.

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
