# Peer review of "Silymarin Inhibits Glutamate Release and Prevents against Kainic Acid-Induced Excitotoxic Injury in Rats"

_biomedicines, 2020, doi:10.3390/biomedicines8110486_

Round 1
Reviewer 1 Report
The paper describes a well planned, well conducted and focused work aimed in demonstrating that the protective action of silymarin against glutamate excitotoxicity acts through ERK signaling. A combined in vitro / in vivo approach has been used. Minor improvements are required, especially to the description of the results. A conclusive analysis to better link the in vitro to the in vivo data has been suggested. In detail:
Title. Because silymarin was injected before inducing KA damage, in the title, the word “protects” has to be changed in “prevents”.
Abstact. As currently described, the results are too much complicated due to the excessive detail on the compound used. Please, simplify the text by limiting it to the description of the biological effects elicited by the inhibitors (without mentioning them).
Some improvement is required to the material and methods section.
- Section 2.1: Add the biological effect of the inhibitors/inducers used to this section. This will help the readers to immediately understanding the rational of your work and better following the description of the results. A recap may be left in the result section, as it is actually.
- Still section 2.1. Some chemical used in the work is missing (e.g: H89, GF109203X; PD98059; GF109203X; FR180204). Please check carefully and add it (with some words about the molecular function) to the section.
- Section 2.3 Synaptosomes preparation. Why did you prepared the synaptosomes for the in vitro part starting from the cerebral cortex, while in the in vivo experiments the hippocampus was analyzed? Why you did not prepare the synaptosomes from dissected hippocampi in order to maintain a perfect parallel between the in vitro and in vivo models?
- Section 2.7. Please modify the title accordingly to the content of the section. We suggest “Animal procedures and histological analysis…”.
Results section.
In general, it is difficult to follow the description of the results because it is difficult to orientating between the text and the figure, as well as to follow the reasoning and interpretation of the experiments. It is suggested to immediately indicating the panel of the figure that will subsequently described at the very beginning of each sentence throughout the text, as well as to write again the description of the results adding some comment about their interpretation. This will help the readies in following your excellent but complex experiments.
- Figure 1B. It looks like silymarin reach a plateau in its inhibitory effect at 30uM. As reported by The Authors, at that point a residual release of glutamate is still present. This suggest the presence of additional pathways, not influenced by silymarin, controlling the neurotransmitter release. Please add some comment this point to the result section.
- Section 3.1 and Figure 1C. The description of the results has to be written again, because not clear as it is. Please add the statistical analysis comparing the “Ca free/EDA +/-silymarin” and Bafilomycin +/- Silymarin” groups to the Silymarin group. Some clearer interpretation of the data has to be added to the section. Have different (lower) concentrations of Ca-free/EDTA and Bafilomycin testes? Is it possible that the concentration of the two inhibitors are too high for allowing recognizing a possible additive effect of silymarin?
- Section 3.2 and Figure 2. Please add some clearer interpretation of the results concerning the inhibitors +/- silymarin challenging to the results section (e.g: “The absence of additive effect of silymarin in respect to gamma-CgTXMVIIC alone, suggested that …”, “while the additive effect of silymarin to CGP37157, indicated that…...”. Actually, this information is resumed on the title of the section, but this is not enough to allow the readers to follow easily your work. Please add your reasoning to the text describing the results.
- Section 3.4. As before asked, please add the inhibitor to the 2.1 sections.
- Section 3.4 and figure 4. Please improve the text by clearly describing and interpreting the results of Figure 4. Please add the statistical analysis of each treatment versus silymarin alone.
- Figure 6 and its legend. Remove the letters (A, A1, B, B1, etc.) from the pictures; they are not necessary and even not corresponding to the legend to the figure. Please add “Hip” on the top of the firs column of picture. Change E with B for the “Quantification of the number of the living neurons”. There is some trouble with the footnote to Figure 6B: please correct the presence/absence of KA/silymarin 50/100 addition. Apparently, all of them have been added (+) everywhere in column 2-4.
- Figure 7 and its legend. Same comments than in the Figure 6.
- Figure 8. It would be great to quantify p-ERK1/2 and ERK1/2 in the hippocampi samples. I understand that the hippocampi represent a complex and heterogeneous tissue, different from a pure synaptosomes preparation. Nevertheless, if an increase in the p-ERK level will be observed, this will strongly connect the in vitro to the in vivo data, supporting or even improving the message of the paper. Due to the relevant damage induced by KA in vivo (Fig 6 and 7), it is possible that the modulation of the level of phosphorylation in the hippocampi would be sufficient to be detected by Western blot analysis on whole hippocampi homogenates.
Discussion.
- Page 13 line 19 and everywhere in the text. Because silymarin was injected before inducing the damage by KA, please modify everywhere in the text the term “protecting / neuroprotective/ etc” with “preventing /preventive of the neurological damage”.
- Page 13, line 21-22. Basal glutamate release. Would be important to quantify glutamate brain (hippocampi)content in animals exposed to “KA + silymarin” and “KA alone” compared to untreated animals (controls) to additionally prove the effect of silymarin in vivo and strength the parallel with the in vitro data.
- Page 13, line 27. In order to improve the readability of the Ms, please, replace VDCCs with the full-length definition.
- Page 15, lines 18-19. The sentence looks confounding. Based on your data, 50-100 mg/Kg are effective in preventing KA induced damage in rodents. please clarify or remove the sentence.
Additional question
Did you quantify the silymarin concentration in the hippocampi of the animals?
Author Response
We thank the reviewers for the critical comments and constructive suggestions.
Reviewer 1
The paper describes a well planned, well conducted and focused work aimed in demonstrating that the protective action of silymarin against glutamate excitotoxicity acts through ERK signaling. A combined in vitro / in vivo approach has been used. Minor improvements are required, especially to the description of the results. A conclusive analysis to better link the in vitro to the in vivo data has been suggested. In detail:
Title. Because silymarin was injected before inducing KA damage, in the title, the word “protects” has to be changed in “prevents”.
Response 1: As suggestion by the reviewer, the word “protects” has to be changed in “prevents”.
Abstact. As currently described, the results are too much complicated due to the excessive detail on the compound used. Please, simplify the text by limiting it to the description of the biological effects elicited by the inhibitors (without mentioning them).
Response 2: As suggestion by the reviewer, several sentences in the abstract have been modified (Page 1, Lines 29-32).
Some improvement is required to the material and methods section.
Section 2.1: Add the biological effect of the inhibitors/inducers used to this section. This will help the readers to immediately understanding the rational of your work and better following the description of the results. A recap may be left in the result section, as it is actually.
Response 3: As suggestion by the reviewer, the biological effect of the inhibitors are added in the section 2.1 (Page 2, Lines 25-39).
Still section 2.1. Some chemical used in the work is missing (e.g: H89, GF109203X; PD98059; GF109203X; FR180204). Please check carefully and add it (with some words about the molecular function) to the section.
Response 4: As suggestion by the reviewer, H89, GF109203X, PD98059, GF109203X and FR180204 are added in the section 2.1 (Page 2, Lines 31-35).
Section 2.3 Synaptosomes preparation. Why did you prepared the synaptosomes for the in vitro part starting from the cerebral cortex, while in the in vivo experiments the hippocampus was analyzed? Why you did not prepare the synaptosomes from dissected hippocampi in order to maintain a perfect parallel between the in vitro and in vivo models?
Response 5: In fact, the effect of silymarin on glutamate release in cerebral cortex nerve endings is our initial research interest. With cortical synaptosomal data, we use the KA model to further prove the neuroprotective effect of silymarin. Thank you very much for the suggestion of reviewer. We will pay attention to this point in the future work.
Section 2.7. Please modify the title accordingly to the content of the section. We suggest “Animal procedures and histological analysis…”.
Response 6: As suggestion by the reviewer, the section title is modified to“Animal procedures and histological analysis of neuronal death” (Page 3, Line 38).
Results section.
In general, it is difficult to follow the description of the results because it is difficult to orientating between the text and the figure, as well as to follow the reasoning and interpretation of the experiments. It is suggested to immediately indicating the panel of the figure that will subsequently described at the very beginning of each sentence throughout the text, as well as to write again the description of the results adding some comment about their interpretation. This will help the readies in following your excellent but complex experiments.
Figure 1B. It looks like silymarin reach a plateau in its inhibitory effect at 30uM. As reported by The Authors, at that point a residual release of glutamate is still present. This suggest the presence of additional pathways, not influenced by silymarin, controlling the neurotransmitter release. Please add some comment this point to the result section.
Response 7: Regarding to this point, several sentences “Silymarin did not cause a complete blockade of release even at the highest concentrations used (50 µM) (Figure 1B). Given the robust depression of glutamate release that was seen with 30 mM silymarin, this concentration of silymarin was used in subsequent experiments to investigate the mechanisms that underlie the ability of silymarin to decrease glutamate release” are added in the result section (Page 5, Line 6-10).
Section 3.1 and Figure 1C. The description of the results has to be written again, because not clear as it is. Please add the statistical analysis comparing the “Ca free/EDA +/-silymarin” and Bafilomycin +/- Silymarin” groups to the Silymarin group. Some clearer interpretation of the data has to be added to the section. Have different (lower) concentrations of Ca-free/EDTA and Bafilomycin testes? Is it possible that the concentration of the two inhibitors are too high for allowing recognizing a possible additive effect of silymarin?
Response 8: In order to make the statement of the sentence more clear, the sentence in the result section is modified (Page 5, Lines 10-21). In addition, we did not examine different (lower) concentrations of Ca-free/EDTA and bafilomycin. The dosages of these agents were chosen based on previous experiments of our group and others (Page 2, Lines 39-40).
Section 3.2 and Figure 2. Please add some clearer interpretation of the results concerning the inhibitors +/- silymarin challenging to the results section (e.g: “The absence of additive effect of silymarin in respect to gamma-CgTXMVIIC alone, suggested that …”, “while the additive effect of silymarin to CGP37157, indicated that…...”. Actually, this information is resumed on the title of the section, but this is not enough to allow the readers to follow easily your work. Please add your reasoning to the text describing the results.
Response 9: Several sentences “The release measured in the presence of ω-CgTX MVIIC and silymarin was similar to that obtained in the presence of ω-CgTX MVIIC alone; The lack of additivity in the inhibitory actions of silymarin and ω-CgTX MVIIC on glutamate release, suggested that there is a preferential interaction between the pathway mediated by silymarin and N- and P/Q-type Ca2+ channels. The additivity in the actions of silymarin and dantrolene or CGP37157on glutamate release ,indicated that the action of silymarin is not due to a reduction of Ca2+ release from intracellular stores. ” are added in the result section (Page 6, Line 14-18, 24-28).
Section 3.4. As before asked, please add the inhibitor to the 2.1 sections.
Response 10: As suggestion by the reviewer, H89, GF109203X, PD98059, GF109203X and FR180204 are added in the section 2.1 (Page 2, Lines 31-35).
Section 3.4 and figure 4. Please improve the text by clearly describing and interpreting the results of Figure 4. Please add the statistical analysis of each treatment versus silymarin alone.
Response 11: In order to make the statement of the sentence more clear, several sentences in the result section is modified (Page 8, Lines 3-4, 5-7, 10-13).
Figure 6 and its legend. Remove the letters (A, A1, B, B1, etc.) from the pictures; they are not necessary and even not corresponding to the legend to the figure. Please add “Hip” on the top of the firs column of picture. Change E with B for the “Quantification of the number of the living neurons”. There is some trouble with the footnote to Figure 6B: please correct the presence/absence of KA/silymarin 50/100 addition. Apparently, all of them have been added (+) everywhere in column 2-4.
Response 12: As suggestion by the reviewer, Figure 6 and it legend is modified (Page 11, Lines 1-6).
Figure 7 and its legend. Same comments than in the Figure 6.
Response 13: As suggestion by the reviewer, Figure 7 and it legend is modified (Page 12, Lines 1-5).
Figure 8. It would be great to quantify p-ERK1/2 and ERK1/2 in the hippocampi samples. I understand that the hippocampi represent a complex and heterogeneous tissue, different from a pure synaptosomes preparation. Nevertheless, if an increase in the p-ERK level will be observed, this will strongly connect the in vitro to the in vivo data, supporting or even improving the message of the paper. Due to the relevant damage induced by KA in vivo (Fig 6 and 7), it is possible that the modulation of the level of phosphorylation in the hippocampi would be sufficient to be detected by Western blot analysis on whole hippocampi homogenates.
Response 14: We agree this point mentioned by the reviewer. However, this part of the experiment cannot be performed in the present study due to the limited response time. Hope you can make allowances for this. Thank you very much for the suggestion of reviewer. We will pay attention to this point in the future work.
Discussion.
Page 13 line 19 and everywhere in the text. Because silymarin was injected before inducing the damage by KA, please modify everywhere in the text the term “protecting / neuroprotective/ etc” with “preventing /preventive of the neurological damage”.
Response 15: As suggestion by the reviewer, the word “protect” is changed to prevent ( Page 14, Line 37; Page 15, Line 7, 28).
Page 13, line 21-22. Basal glutamate release. Would be important to quantify glutamate brain (hippocampi)content in animals exposed to “KA + silymarin” and “KA alone” compared to untreated animals (controls) to additionally prove the effect of silymarin in vivo and strength the parallel with the in vitro data.
Response 16: We examined the effect of silymarin on glutamate concentration in KA-treated rats. The data is added in the result section and expressed in Figure 6 (Page 9, Lines 10-13; Page 10, Lines 2-5). Several sentences are added in the method and discussion section (Page 4, Lines 19-24; Page 14, Lines 40-42).
Page 13, line 27. In order to improve the readability of the Ms, please, replace VDCCs with the full-length definition.
Response 17: The word “VDCCs” is changed to voltage-dependent Ca2+ channels (Page 13, Line 30; Page 14, Line 11).
Page 15, lines 18-19. The sentence looks confounding. Based on your data, 50-100 mg/Kg are effective in preventing KA induced damage in rodents. please clarify or remove the sentence.
Response 18: The sentence is modified to “The dose of silymarin used to prevent neuronal damage in our in vivo study is 50-100 mg/kg” (Page 14, Lines 22-23).
Additional question
Did you quantify the silymarin concentration in the hippocampi of the animals?
Response 19: No. Thank you very much for the suggestion of reviewer. We will pay attention to this point in the future work.
The manuscript is edited by the Wallace Academic Editing.

Reviewer 2 Report
Cheng-Wei Lu et al investigate neuroprotective the impact of silymarin on glutamate release and effect on a rat model of 25 kainic acid (KA)-induced excitotoxicity.
The authors found that decreased glutamate release by silymarin, and its dependency upon N- and P/Q-type Ca2+ channels mediated extracellular Ca2+.
I think the overall experiments are well-designed, but it would have been nice to show these data in better resolutions in Figures.
All the Figures are compressed, losing enough pixel resolutions.
Otherwise, I am ok with the current version.
Author Response
We thank the reviewers for the critical comments and constructive suggestions.
Reviewer 2
Cheng-Wei Lu et al investigate neuroprotective the impact of silymarin on glutamate release and effect on a rat model of 25 kainic acid (KA)-induced excitotoxicity.
The authors found that decreased glutamate release by silymarin, and its dependency upon N- and P/Q-type Ca2+ channels mediated extracellular Ca2+.
I think the overall experiments are well-designed, but it would have been nice to show these data in better resolutions in Figures.
All the Figures are compressed, losing enough pixel resolutions.
Otherwise, I am ok with the current version.
Response 1: To make the picture clear, we have enlarged the picture.

Reviewer 3 Report
The Article titled “Silymarin Inhibits Glutamate Release and Protects against Kainic Acid-Induced Excitotoxic Injury in Rats” offered interesting insight on how Silymarin showed neuroprotective effects. The research is not novel as t repeats a previous study by Kim et al, "Beneficial Effects of Silibinin Against Kainic Acid-induced Neurotoxicity in the Hippocampus in vivo"
There are some concerns that need t to be answered:
Kianic acid treatment would induce repetitive seizures and the authors should measure and assess the therapeutic effect of lessening the occurrence of these seizures.
Neuroinflammation should be evaluated for astrocyte and glial cell along with inflammator markers should be assessed as well (IL-1, IL-6 and IL10).
-the excitotoxicity would lead to neuronal injury where protein degradation is also recorded, the approach used by FlouroJade B is not sufficient, NeuN, and apoptotic/necrotic markers by IF and IHC should be evaluated.
The article doesn't mention how many cohorts of animals used and should have a timeline describing these experiments
full blots are needed to be presented and triplicate of each western should be provided as raw data.
The representative images in IF requires DAPI/Hoechst staining for better differentiating between cell staining and artifacts.
Author Response
We thank the reviewers for the critical comments and constructive suggestions.
Reviewer 3
The Article titled “Silymarin Inhibits Glutamate Release and Protects against Kainic Acid-Induced Excitotoxic Injury in Rats” offered interesting insight on how Silymarin showed neuroprotective effects. The research is not novel as t repeats a previous study by Kim et al, "Beneficial Effects of Silibinin Against Kainic Acid-induced Neurotoxicity in the Hippocampus in vivo"
There are some concerns that need t to be answered:
Kianic acid treatment would induce repetitive seizures and the authors should measure and assess the therapeutic effect of lessening the occurrence of these seizures.
Response 1: The effect of silymarin on KA-induced seizures is added in the result section and expressed in Figure 6 (Page 9, Lines 5-9; Page 10, Lines 2-5).
Neuroinflammation should be evaluated for astrocyte and glial cell along with inflammator markers should be assessed as well (IL-1, IL-6 and IL10).
Response 2: We agree this point mentioned by the reviewer. However, this part of the experiment is not performed in the present study due to the antiinflammation of silymarin has been studied in numerous experiments. Hope you can make allowances for this.
-the excitotoxicity would lead to neuronal injury where protein degradation is also recorded, the approach used by FlouroJade B is not sufficient, NeuN, and apoptotic/necrotic markers by IF and IHC should be evaluated.
Response 3: Thank you very much for the suggestion of reviewer. We will pay attention to this point in the future work. Hope you can make allowances for this.
The article doesn't mention how many cohorts of animals used and should have a timeline describing these experiments
Response 4: The number of animals (n) is added in the figure legend (Page 3, Line 46; Page 6, Line 8; Page 7, Line 5, 21; Page 8, Line 23; Page 9, Line 3; Page 10, Line 4; Page 11, Line 5; Page 12, Line 3; Page 13, Line 15). In addition, the experimental design is shown in Figure 6A (Page 3, Lines 40-41; Page 10, Line 2).
full blots are needed to be presented and triplicate of each western should be provided as raw data.
Response 5: Raw data of western blot is provided during submission.
The representative images in IF requires DAPI/Hoechst staining for better differentiating between cell staining and artifacts.
Response 6: Thank you very much for the suggestion of reviewer. We will pay attention to this point in the future work. Hope you can make allowances for this.
The manuscript is edited by the Wallace Academic Editing.
Round 2
Reviewer 3 Report
The neuroinflammation shoukd be performed along with the dapi staining of the if images.
Also neural injury should be evaluated by studying protein degradation.
Cohort numbers shoukdeshoukd be in the material and method not figure s.
None of the requested comments were fixed.
Author Response
Response to reviewers
Biomedicines-967628R2
We thank the reviewer for the critical comments and constructive suggestions.
Reviewer 3
The neuroinflammation shoukd be performed along with the dapi staining of the if images.
The effect of silymarin on KA-induced microglia and astrocyte activation is added in the result section and expressed in Figure 9 and 10 (Page 12, Lines 5-10; Page 13, Lines 1-10; Page 14, Lines 1-9). In addition, several sentences are added in the discussion section (Page 17, Lines 1-14)
Also neural injury should be evaluated by studying protein degradation.
Neuronal damage in a rat model of excitotoxicity induced by KA was assessed by NeuN staining. Result is added in the result section and expressed in Figure 8 (Page 10, Lines 12-19; Page 12, Lines 1-4).
Cohort numbers shoukdeshoukd be in the material and method not figure s.
The number of animals (n) is added in the material and method section (Page 2, Lines 42-43).
Round 3
Reviewer 3 Report
Dear Authors thank you for the response
please provide full Blots of the Western blotting Data and also please show the triplicate data of each Western blotting
Author Response
Response to reviewers
Biomedicines-967628R3
We thank the reviewer for the critical comments and constructive suggestions.
Reviewer 3
please provide full Blots of the Western blotting Data and also please show the triplicate data of each Western blotting
During the experiment, the molecular weight we wanted was retained, and the rest were cut off. So we cannot provide the full blots of the western blotting data. We are very grateful for the reviewer's suggestion. We will keep all the data for this part of the experiment in the future. The triplicate data of each Western blotting are presented as follows.
